# CoT-RVS: Zero-Shot Chain-of-Thought Reasoning Segmentation for Videos

**Shiu-hong Kao**
The Hong Kong University of Science and Technology,
National University of Singapore
shkao@u.nus.edu

**Yu-Wing Tai**
Dartmouth College
yu-wing.tai@dartmouth.edu

**Chi-Keung Tang**
The Hong Kong University of Science and Technology
cktang@cs.ust.hk

## Abstract

Reasoning Video Object Segmentation is a challenging task, aiming at generating a mask sequence from an input video given a complex and implicit text query. While existing works finetune Multimodal Large Language Models (MLLM) for the task, they still fail in video inputs given complex temporally-sensitive queries, indicating their lack of temporal and spatial integration in complex scenarios. In this paper, we propose **CoT-RVS**, a novel framework employing the zero-shot Chain-of-Thought (CoT) capability of MLLM to address these complex challenges by **temporal-semantic reasoning**: CoT-RVS analyzes the visible objects within a given frame that possibly match the language query (semantic), and chooses a corresponding keyframe for each object that can be observed effortlessly among all frames (temporal). Notably, the CoT-RVS framework is training-free and compatible with closed-source MLLMs, which can be applied to Reasoning Video Instance Segmentation. Our framework's training-free feature further allows its extension to process online video streams, where the CoT is used at test time to update the object of interest when a better target starts to emerge and becomes visible. We conduct extensive experiments on video object segmentation with explicit and implicit queries. The results show that CoT-RVS significantly outperforms previous works in both cases, qualitatively and quantitatively.

## 1 Introduction

Reasoning Video Object Segmentation (Reasoning VOS) presents a formidable challenge in the field of computer vision, requiring the generation of a mask sequence from an input video alongside an implicit and often complex text query (Yan et al., 2024; Bai et al., 2024). Unlike traditional vision-language tasks that directly associate visual data with textual descriptions, Reasoning VOS requires more advanced cognitive capabilities due to the dynamic nature of video data, where factors such as temporally sensitive queries, occlusions and disocclusions due to rapidly moving object can complicate the segmentation process.

Existing approaches of reasoning segmentation focus on finetuning the multimodal Large Language Models (MLLMs) for segmentation-based tokens. While achieving good results in Reasoning VOS, the finetuning process is time consuming, and they still suffer from low accuracy when temporal reasoning is required. An implicit and time sensitive query, *"Which player makes a three-point shot in this basketball game?"*, for instance, is more challenging than *"Which player is wearing black?"* (Fig. 1). A query is *time* or *temporally sensitive* if the relevant video objects *with the specified action context* (*e.g.,* three-point shot), do not appear uniformly over time but rather are concentrated at certain intervals, requiring high-level temporal reasoning to correctly find them to the satisfaction of the query requirement. Even though reasoning segmentation has achieved impressive performance on images (Lai et al., 2023; Kao et al., 2025; Liu et al., 2025) and videos (Bai et al., 2024), current results are far from perfect in the video domain, showcasing failure in integrating temporal information with spatial and textual context in satisfying complex and time-sensitive queries.

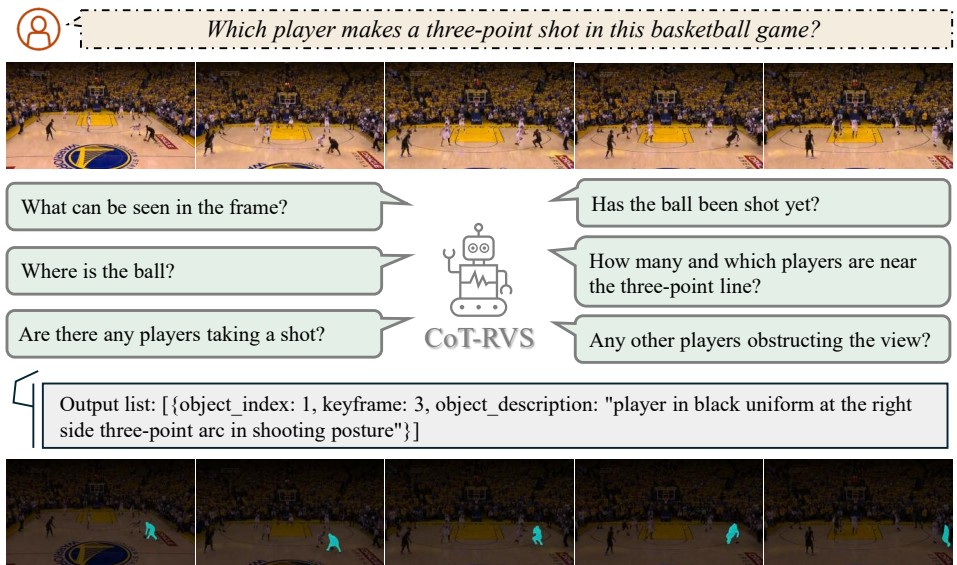

Figure 1: **CoT-RVS** is a novel framework for Reasoning Video Segmentation utilizing the zero-shot Chain-of-Thought (CoT) capability of pretrained multimodal Large Language Models, and correctly segments the fast-moving player right before, during, and after his three-point shot after CoT, given a temporally sensitive query "*Which player makes a three-point shot in this basketball game?*" which requires both spatial and temporal reasoning.

To address this challenge, this paper aims at injecting stronger temporal reasoning ability into a segmentation framework through diversely pre-trained MLLMs empowered by zero-shot Chain-of-Thoughts (Wei et al., 2022). We enable new model capabilities through carefully designed task-specific prompts and innovative module interaction, a visionary trend also shared by Guo et al. (2022); Lian et al. (2023); Hu et al. (2023). Thus, we propose CoT-RVS, a novel, modular framework, whose key success lies in its exceptional reasoning capability on the *temporal-semantic correlation* across keyframe candidates, see an example in Fig. 2. At inference time, CoT-RVS analyzes the visible objects within a given frame that possibly matches the language query (semantic), and chooses a corresponding keyframe for each instance of interest, where the object can be observed with the least effort among all frames (temporal).

Thus, our CoT analysis requires complex, human-level reasoning within and among different video frames, *e.g., "Frame 3 and 4 contains the player in shooting posture, which satisfies the user query, while frame 3 is a better keyframe with a clearer view.".* We will show in this paper that an accurate, in-depth reasoning for this temporal-semantic correlation will significantly improve the model performance in processing time sensitive query.

One of the other standout features of the CoT-RVS framework is its training-free nature, making it compatible with open-source and closed-source MLLMs. We leverage the state-of-the-art Gemma3 (Team et al., 2025) and GPT-4o (Hurst et al., 2024) as the "thinking" agent in our CoT-RVS framework to generate keyframe selectivity. To evaluate the effectiveness of CoT-RVS, we conduct comprehensive experiments on referring VOS datasets (*e.g.,* MeViS (Ding et al., 2023a) and Refer-DAVIS-17 (Pont-Tuset et al., 2017)) and reasoning VOS datasets (*e.g.,* ReVOS (Yan et al., 2024) and ReasonVOS (Bai et al., 2024)). We demonstrate that CoT-RVS significantly outperforms existing referring and reasoning VOS approaches, qualitatively and quantitatively, with its more general formulation based on Reasoning Video Instance Segmentation (Reasoning VIS), which can output multiple instance-level mask sequences to satisfy the user query. In addition, we propose a variant version of CoT-RVS for online video streams, where CoT is used to update the target object when a new object starts to appear in the middle of the video that aligns better with the text query.

In summary, our contributions include:

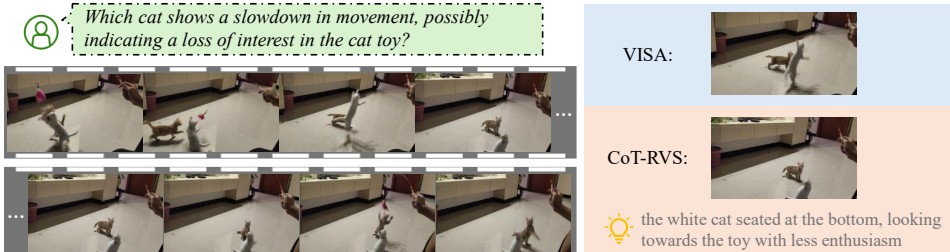

Figure 2: **Keyframe selection is particularly challenging when processing time sensitive queries.** Given a video, where two cats are playing with the cat teaser at the beginning, and then the white cat stays static in the second half of the video, existing methods e.g., VISA (Yan et al., 2024) fail to find a proper keyframe and directly use the user prompt for reasoning segmentation. In contrast, CoT-RVS extracts temporal-semantic correlation from the input video and successfully outputs a more reasonable keyframe with detailed description to the target object.

- A modular, zero-shot reasoning VOS framework that coordinates MLLMs, image segmenters, and video processors in both offline and online settings, without any training or fine-tuning, unlike prior methods.

- A keyframe selection pipeline based on CoT prompting, specifically tailored for temporal reasoning, where MLLMs are required to both localize and describe scene-relevant frames, beyond simple object retrieval.

- An online reasoning extension that adaptively re-selects keyframes at inference time, a feature rarely explored in existing R-VOS systems, enabling streaming video processing.

## 2 RELATED WORK

**Video segmentation.** Video object segmentation (VOS) tracks objects in a given video and output corresponding mask sequences. Early VOS approaches (Maninis et al., 2018; Caelles et al., 2017) rely on online first-frame finetuning, thereby constrained to specific applications because of the high time cost. To speed up inference, follow-up works adopt efficient online learning algorithm (Robinson et al., 2020; Bhat et al., 2020), MRF graph inference (Bao et al., 2018), temporal CNN (Xu et al., 2019; Hou et al., 2019), capsule routing (Duarte et al., 2019), tracking (Jang & Kim, 2017; Wang et al., 2019; Perazzi et al., 2017; Chen et al., 2020), embedding learning (Yang et al., 2020; Voigtlaender et al., 2019; Chen et al., 2018) and space-time matching (Hu et al., 2018; Oh et al., 2019a; Cheng et al., 2021b). Recent VOS methods allow various control modalities, such as categories (Cheng et al., 2021a; Wang et al., 2021; Wu et al., 2022d; Zhang et al., 2023c; 2025; Zhou et al., 2024), points (Ravi et al., 2024; Rajič et al., 2025; Homayounfar et al., 2021), segmentation masks (Cheng & Schwing, 2022; Cheng et al., 2018; Oh et al., 2019b; Perazzi et al., 2017; Ravi et al., 2024), or text query (Botach et al., 2022a; Ding et al., 2023a; Wu et al., 2023b; Seo et al., 2020; Ye et al., 2019; Li et al., 2023b; Wu et al., 2022b; Miao et al., 2023a; Cuttano et al., 2024), making the segmentation more flexible and interactive.

Specifically, building on the advanced development of LLM, Reasoning VOS focuses on generating mask sequences from a highly implicit and complex text query. VISA (Yan et al., 2024) first selects a keyframe and follows the core idea of LISA (Lai et al., 2023) by generating a segmentation-oriented token with finetuned image-based MLLM. Video-LISA (Bai et al., 2024) and HyperSeg (Wei et al., 2024) encode the video input in a multi-grained manner and concatenate the text query, feeding both to the LLM for segmentation-based outputs. However, none of the prior works incorporate adequate consideration of CoT reasoning ability for spatial and temporal information in Reasoning VOS. Our CoT-RVS is the first zero-shot-CoT framework compatible with pre-trained MLLM and image segmentation module, making it flexible to a boarder audience. In fact, recent studies, e.g. (Snell et al., 2024), have emphasized that test-time compute and inference-time reasoning are becoming increasingly central to leveraging pre-trained LLMs. Our method is aligned with this direction.

**Chain-of-Thought MLLM.** Drawing inspiration from the exceptional reasoning capabilities of Large Language Models (LLMs), Multi-modal Large Language Models (MLLMs) seek to integrate visual components to enhance multimodal perception and reasoning. These models are often employed to produce captions or respond to inquiries based on visual and textual inputs, typically achieved by techniques such as prompt engineering (Liu et al., 2023c; Shen et al., 2023; Wu et al., 2023a),

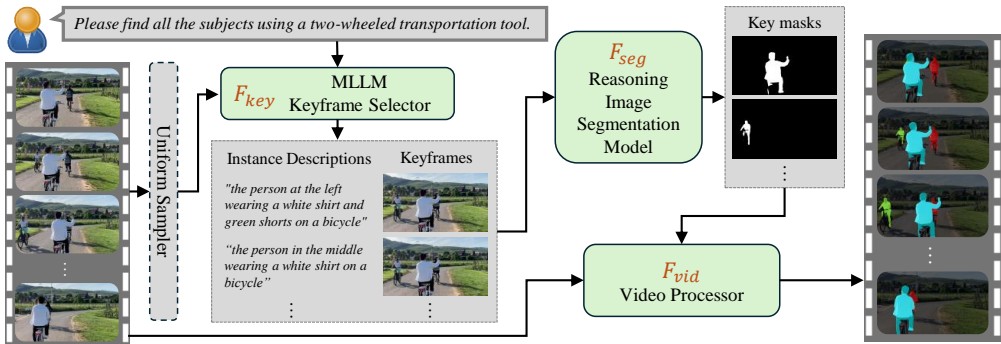

Figure 3: **CoT-RVS for Reasoning VIS** where Reasoning VOS is a special case.

cross-attention mechanism (Alayrac et al., 2022), Q-Former (Dai et al., 2023; Li et al., 2023a), prompt tuning (Zhang et al., 2023b), linear projection (Koh et al., 2023; Liu et al., 2023b), and unified model architectures (Peng et al., 2023; Xiao et al., 2024). These MLLMs have greatly revolutionized the traditional vision-centric tasks like image segmentation (Lai et al., 2023; Xia et al., 2024; Zhang et al., 2023a; He et al., 2024; Bao et al., 2024), object detection (Zhang et al., 2023a; Zang et al., 2024; You et al., 2023; Chen et al., 2023; Ma et al., 2024), video segmentation (Yan et al., 2024; Bai et al., 2024; Wei et al., 2024), and object tracking (Wang et al., 2024; Munasinghe et al., 2023; Zhu et al., 2023).

In addition to these advancements, CoT-based LLMs have been proposed to improve the reasoning capability (Wei et al., 2022; Wang et al., 2022). Existing CoT approaches tend to concentrate on inference through progressive reasoning demonstrations (Zhang et al., 2023d; Wei et al., 2022; Lyu et al., 2023) or by utilizing explicit prompts that clearly outline each step (Kojima et al., 2022). However, enabling CoT capabilities for MLLMs typically undergo fine-tuning on specifically designed multimodal CoT datasets (Mondal et al., 2024; Zhang et al., 2023e; Lu et al., 2022) or incorporate intricate intermediate representations (Mitra et al., 2024; Surís et al., 2023), which restricts their usability and scalability due to data accessibility and time complexity. Closer to our work, recent works, such as ThinkFirst (Kao et al., 2025) and Seg-Zero (Liu et al., 2025), have successfully incorporate CoT MLLM in reasoning image segmentation, while still falling short of "thinking" in the temporal domain. In contrast, we aim to, for the first time, abridge this gap by exploring zero-shot CoT reasoning of pre-trained MLLMs and apply them in the area of video segmentation.

## 3    CoT-RVS: Multi-Agent Framework

CoT-RVS is a multi-agent framework that consists of three network components: an MLLM keyframe selector $\mathcal{F}_{key}$, a reasoning image segmentation model $\mathcal{F}_{seg}$, and a video processor $\mathcal{F}_{vid}$, see Fig. 3.

**Input and Output Settings.** Given an input video $x_v = \left\{ I_i \in \mathbb{R}^{H \times W \times 3} \right\}_{i=1}^{T}$ and a text query $q$ that can be very complex, CoT-RVS is inherently a Reasoning Video Instance Segmentation model $\psi_\theta(\cdot)$ which generates a list of non-overlapping binary mask sequences $\hat{y} = \left\{ \mathcal{M}_i \in \mathbb{R}^{T \times H \times W} \right\}_{i=1}^{k}$:

$$\hat{y} = \psi_\theta(x_v, q), \tag{1}$$

where $k$ indicates the number of object instances of interest that satisfy the requirements of input query $q$, where Reasoning Video Object Segmentation (VOS) is clearly reducible to Reasoning Video Instance Segmentation (VIS) by setting $k = 1$.

Each output mask sequence $\mathcal{M}_i = \{m_{i,t} \in \mathbb{R}^{H \times W}\}_{t=1}^{T}$ corresponds to an object instance of interest with the same number of frames and size as $x_v$, having the property of $m_{i,t} \cap m_{j,t} = \phi$ for $i \neq j$.

**Query requirement.** CoT-RVS shares a similar query setting with the first Reasoning Video Object Segmentation (Yan et al., 2024), where the input query $q$ is complex or implicit (*e.g., "Please find the vehicle in the video which can accommodate at least four people."*), in contrast to the Referring VOS (Seo et al., 2020) task which takes explicit and direct query (*e.g. "Please segment the red car in the video."*).

### 3.1    Multi-Agentic Framework

**MLLM Keyframe selector.** The keyframe selector is an MLLM agent, *e.g.,* GPT-4o (Hurst et al., 2024) or Gemma3 (Team et al., 2025), that generates *keyframe selectivites* from a list of keyframe

candidates, denoted as $\{I_i^c\}$, with a CoT thinking process that analyzes the temporal-semantic correlation among the input video frames. To find all the object instances that meet the query requirement necessitating temporal-semantic understanding, we select their corresponding keyframes such that the pertinent instance is most apparent for the MLLM agent to observe from that frame. Thus, the output contains a list of selected object instances, each of which corresponds to an object index, a respective keyframe, and a keyframe-related object description (*e.g., "the player in black jersey at the right side of the frame"*).

Refer to Appx. A for the prompting details. In implementation, the agent is prompted to automatically synthesize a series of questions for each keyframe candidate and answer them one-by-one. Starting from general semantic information like *"What can be seen in the frame?"*, to temporal-related questions like *"Is it a better keyframe?"*, and eventually ending with detailed questions like *"Are there any new objects that meet the query requirement?"*, this CoT process enables thorough reasoning between the user query and the temporal-semantic correlation among the keyframe candidates.

After analyzing all the keyframe candidates, the agent will collect all the per-frame responses and generate an output list composed of all target instances along with the respective keyframes based on its progressive question-and-answer pairs. We demonstrate this CoT process in Fig. 4 for visualization.

**Reasoning Image Segmentation Model.** The task of the reasoning image segmentation model is to predict the key mask of each selected object instance. This agent is prompted by the selected keyframes and the keyframe-related object descriptions generated from the MLLM keyframe selector.

**Video Processor.** The task of the video processor is to track the selected object instances over the entire video. Each key mask generated by the reasoning image segmentation model will be tracked to produce an instance-level mask sequence.

## 3.2 CoT-RVS: Reasoning Video Instance Segmentation

CoT can be applied to the entire video when available for reasoning instance segmentation. Given an input video $x_v = \left\{ I_i \in \mathbb{R}^{H \times W \times 3} \right\}_{i=1}^{T}$ and a text query $q$, we first uniformly sample a number of keyframe candidates $\{I_i^c \in x_v\}_{i=1}^{T'}$, where $T' = \lfloor T/\xi \rfloor$ for some $\xi < T$. These candidates, along with the user query $q$, are then sent to the MLLM keyframe selector $\mathcal{F}_{key}$ for temporal-semantic analysis, as described in Sec. 3.1, which eventually outputs a keyframe selectivity $S$ composed of a list of instance-level text descriptions $\{s_i\}_{i=1}^{k}$ with respect to their corresponding keyframes $\{f_i\}_{i=1}^{k}$ (*e.g.,* "the red car" in $I_2^c$, "the blue motorcycle" in $I_4^c$, ...). Specifically, $k$ is the predicted number of target instances. This CoT process can be simplified as:

$$I_{1:T'}^c = \text{Sample}\left(I_{1:T}, \xi\right); \tag{2}$$

$$s_{1:k}, f_{1:k} = \mathcal{F}_{key}\left(q, I_{1:T'}^c\right). \tag{3}$$

Following by the extraction of temporal-semantic correlation within a video, where time sensitivity has been resolved, we can now incorporate the image-based segmentation model and video processor to generate temporal-consistent mask sequence. In particular, we use the reasoning image segmentation

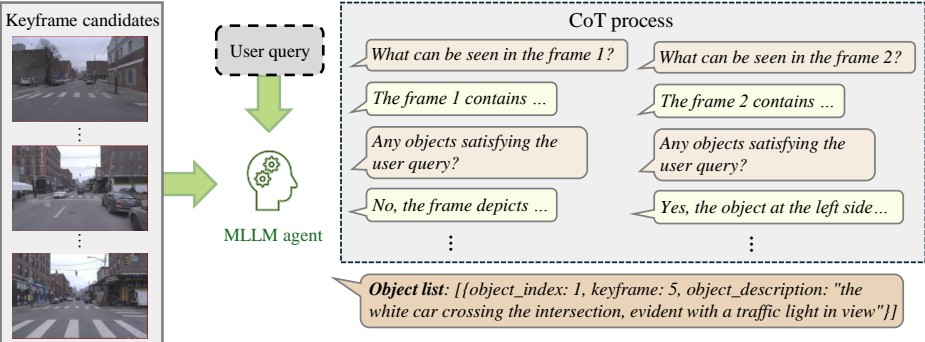

Figure 4: **Illustration of the CoT Process.** The MLLM agent is prompted to synthesize chain-of-thought questions and answers. This CoT process enhances the in-depth temporal and spatial understanding for the keyframe candidates. Based on the CoT result, the MLLM agent outputs an instance-level object list, which contains the object-specific keyframe and description within the frame. Refer to the Appendices. A and E for the detailed prompt and output examples.

model $\mathcal{F}_{seg}$ to generate per-instance key masks $\{\tilde{m}_i \in \mathbb{R}^{H \times W}\}$ based on the the instance-level selectivity. These key masks will be sent to the video processor $\mathcal{F}_{vid}$, *i.e.* SAM2 (Ravi et al., 2024), to predict the per-instance mask sequences. We formulate this process as:

$$\tilde{m}_i = \mathcal{F}_{seg}(s_i, f_i); \quad \hat{\mathcal{M}}_i = \mathcal{F}_{vid}(\tilde{m}_i, x_v) \quad \text{for} \quad i = 1, 2, ..., k. \tag{4}$$

To ensure the mask sequences are not overlapping, we adopt an optional post-processing operation:

$$m_{1,t} = \hat{m}_{1,t}; \tag{5}$$

$$m_{i,t} = \bigcap_{j=1}^{i-1} \neg m_{j,t} \cap \hat{m}_{i,t}, \tag{6}$$

for any $t \leq T$ and $i \leq k$, where the final output $\hat{y} \in \mathbb{R}^{T \times H \times W \times k}$ is the combination of all $m_{i,t} \in \mathbb{R}^{H \times W}$.

### 3.3 CoT-RVS: Online Reasoning Video Object Segmentation

Our approach can be extended to handle online video streams where future frames have yet to be observed. Refer to Fig. 5. The goal of Online Reasoning Video Object Segmentation (Online Reasoning VOS) is to generate a binary mask sequence $\hat{y} = \mathcal{M} = \{m_t \in \mathbb{R}^{H \times W}\}_{t=1}^{T}$ from an online input video stream $\{I_i \in \mathbb{R}^{H \times W \times 3}\}_{i=1}^{T}$, and text query $q$, while the model receives video frames in temporal order and makes predictions only based on frames that have already been seen. Specifically, this process can be modified from Eq. (1) as:

$$m_t = \psi_\theta(I_{1:t}, m_{1:t-1}, q). \tag{7}$$

The final output $\hat{y}$ can be obtained by combining predictions at all time, *i.e.,* $m_1, m_2, ..., m_T$.

The CoT-RVS framework in Online ReasoningVOS version shares a similar architecture to the original ReasoningVIS, while "asking" the keyframe selector $\mathcal{F}_{key}$ for periodic updates of the key mask. Specifically, we introduce a hyper-parameter $\xi$ for the temporal frequency of keyframe selection. We denote $I_t^{key}$ as the selected keyframe at time $t$, and $p_t$ as its corresponding frame index, respectively. When given a complex text query $q$ and an online video frame $I_t$, where $t = n\xi + 1$ for some $n \in \mathbb{Z}$, the model sends them to $\mathcal{F}_{key}$ to justify the the keyframe selectivity $S_t \in \{0, 1\}$, which indicates whether $I_t$ should be chosen as a keyframe. This process can be formulated as:

$$I_0^{key} = \texttt{Null}; \quad p_0 = 0; \tag{8}$$

$$S_t = \mathcal{F}_{key}(q, I_t); \tag{9}$$

$$I_t^{key} = S_t \cdot I_t + (1 - S_t) \cdot I_{\max(t-\xi, 0)}^{key}; \tag{10}$$

$$p_t = S_t \cdot t + (1 - S_t) \cdot p_{\max(t-\xi, 0)}, \tag{11}$$

for $t = 1, \xi + 1, 2\xi + 1, ....$ Refer to Appx. B for the prompting details for Eq. (9). The selected keyframe $I_t^{key}$ is then used to generate a mask control and guide the video processor for $\xi$ consecutive frames. We formulate this process as the following: $m_t = \mathbf{0}$ if $I_{t'}^{key} = \texttt{Null}$; otherwise,

$$\tilde{m}_{t'} = \mathcal{F}_{seg}\left(q, I_{t'}^{key}\right); m_t = \mathcal{F}_{vid}\left(\tilde{m}_{t'}, I_{t':t}\right), \tag{12}$$

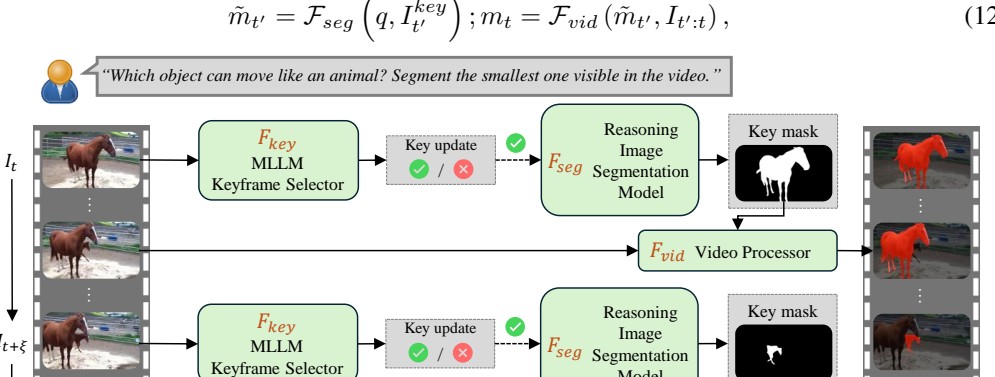

Figure 5: **CoT-RVS for Online Reasoning Video Object Segmentation.**

*"My friend and I each drove our cars to another city. He was driving a white car and leading the way in front of me, but he drove too fast and I lost him. He called me to say that he had just been waiting at a traffic light and then crossed an intersection. Which one is most likely to be my friend's car?"*

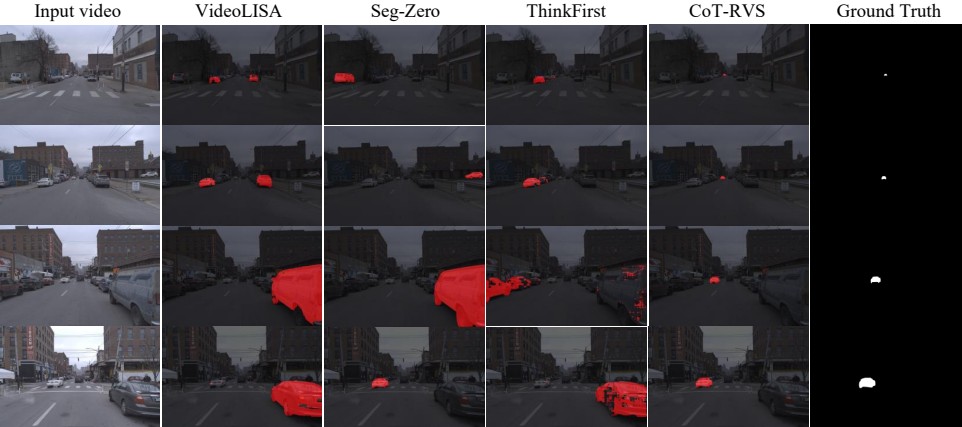

Figure 6: **Qualitative results.** By thinking with CoT, our CoT-RVS demonstrates the temporal reasoning ability previous works fail to achieve. We compare CoT-RVS with Reasoning VOS method VideoLISA (Bai et al., 2024) and CoT Reasoning Image Segmentation methods Seg-Zero (Liu et al., 2025) and ThinkFirst (Kao et al., 2025).

Table 1: Referring VOS results on MeViS benchmark (Ding et al., 2023a).

| Methods | Online | $\mathcal{J}\&\mathcal{F}$ | $\mathcal{J}$ | $\mathcal{F}$ |
|---|---|---|---|---|
| *Traditional methods without reasoning ability* | | | | |
| URVOS (Seo et al., 2020) | ✓ | 27.8 | 25.7 | 29.9 |
| LBDT (Ding et al., 2022) | ✓ | 29.3 | 27.8 | 30.8 |
| MTTR (Botach et al., 2022b) | ✗ | 30.0 | 28.8 | 31.2 |
| ReferFormer (Wu et al., 2022c) | ✗ | 31.0 | 29.8 | 32.2 |
| VLT+TC (Ding et al., 2021) | ✗ | 35.5 | 33.6 | 37.3 |
| LMPM (Ding et al., 2023a) | ✗ | 37.2 | 34.2 | 40.2 |
| *LLM-based methods with reasoning ability* | | | | |
| LISA-7B (Lai et al., 2023) | ✓ | 37.2 | 35.1 | 39.4 |
| LISA-13B (Lai et al., 2023) | ✓ | 37.9 | 35.8 | 40.0 |
| TrackGPT-7B (Zhu et al., 2023) | ✓ | 40.1 | 37.6 | 42.6 |
| TrackGPT-13B (Zhu et al., 2023) | ✓ | 41.2 | 39.2 | 43.1 |
| VideoLISA (OTSA)* (Bai et al., 2024) | ✗ | 42.3 | 39.4 | 45.2 |
| VideoLISA (Po)* (Bai et al., 2024) | ✗ | 44.4 | 41.3 | 47.6 |
| VISA-7B (Yan et al., 2024) | ✗ | 43.5 | 40.7 | 46.3 |
| VISA-13B (Yan et al., 2024) | ✗ | 44.5 | 41.8 | 47.1 |
| SAMWISE (Cuttano et al., 2024) | ✓ | 49.5 | 46.6 | 52.4 |
| GLUS (Lin et al., 2025) | ✗ | 51.3 | 48.5 | 54.2 |
| CoT-RVS-LLaVA1.5-7B | ✓ | 45.9 | 42.7 | 49.1 |
| CoT-RVS-Gemma3-12B | ✗ | 44.2 | 40.3 | 48.1 |
| CoT-RVS-GPT-4o | ✗ | **52.2** | **48.7** | **55.7** |
| * OTSA: One-Token-Seg-All; Po: post-optimization. | | | | |

Table 2: Referring VOS results on Refer-DAVIS-17 benchmark (Pont-Tuset et al., 2017).

| Methods | Online | $\mathcal{J}\&\mathcal{F}$ | $\mathcal{J}$ | $\mathcal{F}$ |
|---|---|---|---|---|
| *Traditional methods without reasoning ability* | | | | |
| URVOS (Seo et al., 2020) | ✓ | 51.6 | 47.2 | 55.9 |
| YOFO (Li et al., 2022) | ✓ | 53.3 | 48.8 | 57.8 |
| MLSA (Wu et al., 2022a) | ✗ | 57.9 | 53.8 | 62.0 |
| ReferFormer (Wu et al., 2022c) | ✗ | 61.1 | 58.1 | 64.1 |
| SgMg (Miao et al., 2023b) | ✗ | 63.3 | 60.6 | 66.0 |
| OnlineRefer (Wu et al., 2023b) | ✗ | 64.8 | 61.6 | 67.7 |
| *LLM-based methods with reasoning ability* | | | | |
| LISA-7B (Lai et al., 2023) | ✓ | 64.8 | 62.2 | 67.3 |
| LISA-13B (Lai et al., 2023) | ✓ | 66.0 | 63.2 | 68.8 |
| TrackGPT-7B (Zhu et al., 2023) | ✓ | 63.2 | 59.4 | 67.0 |
| TrackGPT-13B (Zhu et al., 2023) | ✓ | 66.5 | 62.7 | 70.4 |
| VideoLISA (OTSA)* (Bai et al., 2024) | ✗ | 67.7 | 63.8 | 71.5 |
| VideoLISA (Po)* (Bai et al., 2024) | ✗ | 68.8 | 64.9 | 72.7 |
| VISA-7B (Yan et al., 2024) | ✗ | 69.4 | 66.3 | 72.5 |
| VISA-13B (Yan et al., 2024) | ✗ | 70.4 | 67.0 | 73.8 |
| SAMWISE (Cuttano et al., 2024) | ✓ | 70.6 | 67.4 | 74.5 |
| HyperSeg (Wei et al., 2024) | ✗ | 71.2 | - | - |
| CoT-RVS-LLaVA1.5-7B | ✓ | 73.9 | 70.4 | 77.5 |
| CoT-RVS-Gemma3-12B | ✗ | 74.6 | 70.9 | 78.3 |
| CoT-RVS-GPT-4o | ✗ | **79.1** | **75.3** | **82.9** |
| * OTSA: One-Token-Seg-All; Po: post-optimization. | | | | |

where $t' = p_{\lfloor t/\xi \rfloor \cdot \xi + 1}$ for any $t \leq T$. Specifically, $t'$ implies the frame index of the most recently selected keyframe. This framework adopts a greedy strategy to periodically update the selected keyframe when an incoming frame satisfies the query requirement, then using the keyframe to track in the following frames. When none of the previous frames is selected as a keyframe, the model outputs nothing. Eventually, we obtain the mask sequence $\hat{y}$ with all $m_t$'s.

## 4 EXPERIMENTS

In this section, we will empirically demonstrate the effectiveness of our proposed CoT-RVS. We begin by introducing the experimental settings in Sec. 4.1, followed by showing the qualitative and quantitative results and comparison of CoT-RVS with previous works in Sec. 4.2. Lastly, we conduct ablation study to analyze the keyframe candidates for GPT-4o and CoT thinking process in Sec. 4.3, where more ablation study, *e.g.* robustness, time and latency analysis, will be provided in the appendix.

### 4.1 EXPERIMENTAL SETTINGS

**Implementation details.** We implement our MLLM keyframe selector $\mathcal{F}_{key}$ with GPT-4o (Hurst et al., 2024) (version "2024-05-13") for the original CoT-RVS and LLaVA1.5-7B (Liu et al., 2023a) for the online CoT-RVS. We also conduct experiments with the open-sourced Gemma3-12B (Team

Table 3: Reasoning VOS results on ReasonVOS benchmark (Bai et al., 2024).

| Methods | Online | $\mathcal{J}\&\mathcal{F}$ | $\mathcal{J}$ | $\mathcal{F}$ |
|---|---|---|---|---|
| *Traditional methods without reasoning ability* | | | | |
| MTTR (Botach et al., 2022b) | ✗ | 31.1 | 29.1 | 33.1 |
| ReferFormer (Wu et al., 2022c) | ✗ | 32.9 | 30.2 | 35.6 |
| SOC (Luo et al., 2023) | ✗ | 35.9 | 33.3 | 38.5 |
| SgMg (Miao et al., 2023b) | ✗ | 36.2 | 33.7 | 38.7 |
| OnlineRefer (Wu et al., 2023b) | ✓ | 38.7 | 34.6 | 42.9 |
| *LLM-based methods with reasoning ability* | | | | |
| LISA (Lai et al., 2023) | ✓ | 31.1 | 29.1 | 33.1 |
| VideoLISA (OTSA)* (Bai et al., 2024) | ✗ | 45.1 | 43.1 | 47.1 |
| VideoLISA (Po)* (Bai et al., 2024) | ✗ | 47.5 | 45.1 | 49.9 |
| GLUS (Lin et al., 2025) | ✗ | 49.9 | 47.5 | 52.4 |
| CoT-RVS-LLaVA1.5-7B | ✓ | 52.0 | 49.5 | 54.5 |
| CoT-RVS-Gemma3-12B | ✗ | 50.7 | 47.5 | 54.0 |
| CoT-RVS-GPT-4o | ✗ | **65.5** | **62.4** | **68.7** |

\* OTSA: One-Token-Seg-All; Po: post-optimization.

Table 4: Reasoning VOS results on ReVOS benchmark (Yan et al., 2024).

| Methods | Online | $\mathcal{J}\&\mathcal{F}$ | $\mathcal{J}$ | $\mathcal{F}$ |
|---|---|---|---|---|
| *Traditional methods without reasoning ability* | | | | |
| MTTR (Botach et al., 2022b) | ✗ | 25.5 | 25.1 | 25.9 |
| LMPM (Ding et al., 2023a) | ✗ | 26.4 | 21.2 | 31.7 |
| ReferFormer (Wu et al., 2022c) | ✗ | 28.1 | 26.2 | 29.9 |
| LLaMA-VID (Li et al., 2023c)+LMPM | ✗ | 26.1 | 20.9 | 31.4 |
| *LLM-based methods with reasoning ability* | | | | |
| LISA-13B (Lai et al., 2023) | ✓ | 41.6 | 39.8 | 43.5 |
| TrackGPT-7B (IT)* (Zhu et al., 2023) | ✓ | 43.6 | 41.8 | 45.5 |
| TrackGPT-13B (IT)* (Zhu et al., 2023) | ✓ | 45.0 | 43.2 | 46.8 |
| VISA (Yan et al., 2024) | ✗ | 47.5 | 45.3 | 49.7 |
| VISA (IT)* (Yan et al., 2024) | ✗ | 50.9 | 48.8 | 52.9 |
| GLUS (Lin et al., 2025) | ✗ | 54.9 | 52.4 | 57.3 |
| HyperSeg (Wei et al., 2024) | ✗ | 55.7 | **53.1** | 58.4 |
| CoT-RVS-LLaVA1.5-7B | ✓ | 46.2 | 43.5 | 48.8 |
| CoT-RVS-Gemma3-12B | ✗ | 47.1 | 43.4 | 50.9 |
| CoT-RVS-GPT-4o | ✗ | **55.9** | 52.8 | **59.0** |

\* (IT): instruction tuning with the ReVOS training set.

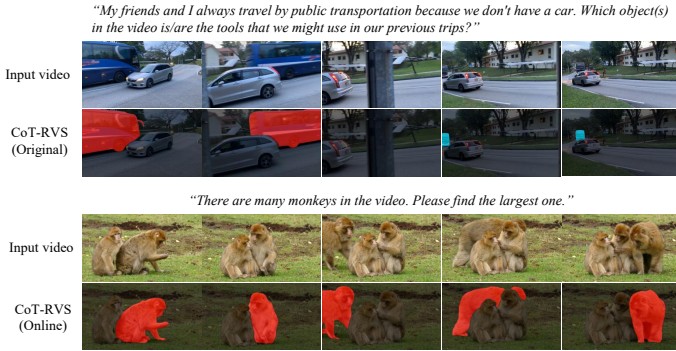

*"My friends and I always travel by public transportation because we don't have a car. Which object(s) in the video is/are the tools that we might use in our previous trips?"*

Input video

CoT-RVS (Original)

*"There are many monkeys in the video. Please find the largest one."*

Input video

CoT-RVS (Online)

Figure 7: **Visual results of the original Reasoning VIS and its online version for Reasoning VOS.** The original CoT-RVS can be used for instance-level reasoning video segmentation, while the online extension is preferred when the object of interest changes during the tracking process. Zoom in for details and refer to the appendix for more examples.

Table 5: Reasoning VOS results with temporally sensitive queries on our sampled T-ReasonVOS dataset.

| Experiments | $\mathcal{J}\&\mathcal{F}$ | $\mathcal{J}$ | $\mathcal{F}$ |
|---|---|---|---|
| LISA | 31.0 | 28.6 | 33.3 |
| ThinkFirst+SAM2 | 33.6 | 31.1 | 36.1 |
| VideoLISA (OTSA) | 37.4 | 35.1 | 39.6 |
| CoT-RVS-Gemma3-12B | 50.2 | 46.8 | 53.6 |
| CoT-RVS-GPT-4o | **55.5** | **51.2** | **59.9** |

Table 6: Ablation study for the GPT-4o input of keyframe candidates on the ReasonVOS benchmark.

| Experiments | $\mathcal{J}\&\mathcal{F}$ | $\mathcal{J}$ | $\mathcal{F}$ |
|---|---|---|---|
| $\xi = \lfloor \frac{T-1}{4} \rfloor + 1$ | 66.2 | 63.1 | 69.4 |
| $\xi = \lfloor \frac{T-1}{8} \rfloor + 1$ | 65.5 | 62.4 | 68.7 |
| $\xi = \lfloor \frac{T-1}{16} \rfloor + 1$ | 60.0 | 57.1 | 62.9 |

et al., 2025) in CoT-RVS for users with limited access to the OpenAI service. Quantitative studies for different MLLMs are provided in Appendices. F and G to analyze our implementation choices. We adopt Seg-Zero (Liu et al., 2025) as the reasoning image segmentation model $\mathcal{F}_{seg}$ and SAM2 (Ravi et al., 2024) as the video processor $\mathcal{F}_{vid}$, where alternatives with LISA (Lai et al., 2023) and Cutie (Cheng et al., 2024) are also demonstrated in Appx. H for modularity analysis. We use the pre-trained weights of each module from their official implementation. We set $\xi = \lfloor \frac{T-1}{8} \rfloor + 1$ for our experiments (*i.e.*, the integer value that generates at most and closest to 8 keyframe candidates), and $\xi = 4$ for online experiments. GPT-4o's temperature is set to 0.5, and the maximum number of output tokens is 2500. To evaluate the VOS task with the VIS version of CoT-RVS, we take the union of all the predicted instance-level mask(s) as the final output.

**Datasets.** We compare CoT-RVS with previous works on multiple benchmarks, including referring VOS datasets (MeViS (Ding et al., 2023a) and Refer-DAVIS-2017 (Pont-Tuset et al., 2017)) and reasoning VOS datasets (ReVOS (Yan et al., 2024) and ReasonVOS (Bai et al., 2024)). To evaluate reasoning VOS with temporally sensitive queries, we manually sample a subset of ReasonVOS, named as T-ReasonVOS, which contains only difficult examples involving time-sensitive queries. Refer to Appx. D for more dataset descriptions.

**Evaluation metrics.** We follow previous works and evaluate the CoT-RVS performance with region similarity ($\mathcal{J}$), contour accuracy ($\mathcal{F}$), and their average value ($\mathcal{J}\&\mathcal{F}$). Specifically, $\mathcal{J}$ evaluates how well the pixels of the predicted mask match the ground truth overall using intersection over union (IoU), and $\mathcal{F}$ combines the precision and recall of boundary pixels, emphasizing the accuracy at the object boundaries.

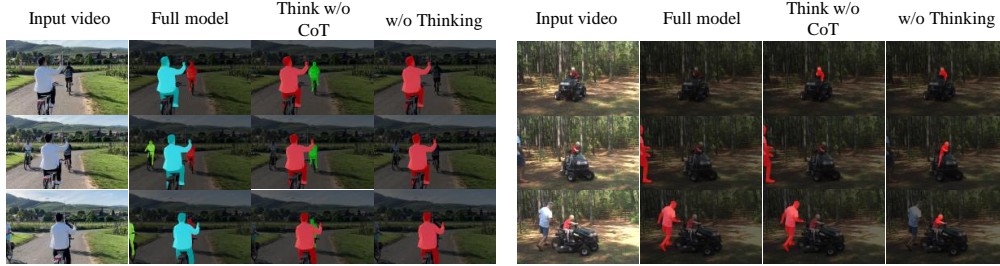

(a) **Reasoning VIS**: *"Please find all the subjects using a two-wheeled transportation tool."*

(b) **Reasoning VOS**: *"When parents see their children in potential danger, they are often concerned and ready to help. Who is the parent in the scene?"*

Figure 8: **Ablation study.** We examine the effectiveness of the Thinking process for keyframe selection. "Think w/o CoT" means selecting keyframes without the CoT-guided prompts. "w/o Thinking" means randomly selecting a keyframe for the original method and selecting the first frame as keyframe for its online version. Zoom in for details.

## 4.2 MAIN RESULTS

**Referring VOS.** We report the quantitative comparisons of CoT-RVS with previous works on the tasks of Referring VOS in Tab. 1 and Tab. 2. Online methods are highlighted in gray in these tables. Results show that both of the online and offline versions of CoT-RVS significantly outperform the state of the art. CoT-RVS achieves up to $3.5\%$ improvements over the online approaches and $15.6\%$ over the offline approaches. Notably, online CoT-RVS surpasses offline baselines in most cases, while given a more challenging task setting.

**Reasoning VOS.** We demonstrate the quantitative comparison of the ReasonVOS benchmark and ReVOS benchmark in Tab. 3 and Tab. 4, respectively, where CoT-RVS achieves up to $18\%$ than state-of-the-art offline methods and up to $13.3\%$ than state-of-the-art online methods, showcasing the better reasoning capability of "thinking" first. Implementation based on Gemma3-12B performs slightly worse results than in GPT-4o, possibly due to the capability of temporal reasoning and image understanding. We further report the qualitative performance of reasoning VOS with temporal-related queries in Fig. 6. When the temporal reasoning capability is needed, CoT-RVS achieves significantly better performance than any existing works. This is also verified in the quantitative results of T-ReasonVOS dataset, as shown in Tab. 5.

**Reasoning VIS and online Reasoning VOS.** We demonstrate the extended features of CoT-RVS in Fig. 7. In specific, the original CoT-RVS is able to generate instance-level mask sequences. The online version of CoT-RVS can update the object of interest during the tracking process. These extensions provide CoT-RVS with more flexibility to broader audience.

## 4.3 ABLATION STUDY

**Analysis of hyper-parameter $\xi$.** We sample a keyframe candidate for every $\xi$ frames, then concatenate the keyframe candidates into a grid image for the GPT-4o CoT process. In Tab. 6, we compare the reasoning VOS results on the ReasonVOS benchmark (Bai et al., 2024) with different $\xi$ values. Our empirical results showcase that GPT-4o achieves comparable performance when $T' \leq 8$ (*i.e.,* $\xi \geq \lfloor \frac{T-1}{8} \rfloor + 1$), while the image reasoning capability drastically decreases with $\xi = \lfloor \frac{T-1}{16} \rfloor + 1$. Therefore, to preserve the temporal information in the video without sacrificing GPT-4o reasoning accuracy, we recommend selecting a $\xi$ value such that $4 \leq T' \leq 8$.

**Analysis of CoT process.** We examine the CoT process of the MLLM keyframe selector. Specifically, we show in Fig. 8 that the CoT process significantly improves the reasoning capability. In Reasoning VIS, thinking without CoT may result in missing some instances that require more effort to find. In Online Reasoning VOS, thinking without CoT degrades the performance of the keyframe justification, possibly leading to wrong classification of binary selectivity. All of these experiments point to the soundness of our proposed CoT reasoning.

**Limitations.** While CoT-RVS demonstrates strong performance in reasoning video segmentation through zero-shot Chain-of-Thought prompting, a key limitation of CoT-RVS is its reliance on pre-trained MLLMs without task-specific fine-tuning. As a result, its performance is bounded by the general reasoning capabilities of these models. We believe that fine-tuning MLLMs for temporally-aware multimodal reasoning could further improve segmentation accuracy.

## 5 CONCLUSION

In this paper, we presented CoT-RVS, a novel zero-shot, training-free framework that leverages Chain-of-Thought (CoT) capabilities of multimodal large language models (MLLMs) to enhance reasoning in video segmentation tasks. Our extensive experiments on various benchmark datasets demonstrate that CoT-RVS significantly outperforms state-of-the-art methods, particularly in challenging scenarios involving complex, implicit and time sensitive queries, occlusions, and rapid object movements. Additionally, we extended the framework for online video streams, showcasing its adaptability for a broader audience. Overall, CoT-RVS sets a new benchmark in reasoning video segmentation and opens avenues for future research in zero-shot reasoning and its applications in dynamic visual environments.

## ACKNOWLEDGMENTS

This work was supported in part by the Research Grant Council of the Hong Kong SAR under Theme-based Research Scheme, grant no. T22-606/23R.

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

## APPENDIX A  COT-RVS COT PROMPT DETAILS

One of the key components of the CoT-RVS framework is the chain-of-thought keyframe selection process with temporal-semantic analysis using GPT-4o and Gemma3 (Eq. (3)). To prompt GPT-4o Hurst et al. (2024) or Gemma3-12B Team et al. (2025), we first resize and concatenate all the keyframe candidates $\{I_i^c\}_{i=1}^{T'}$ into a merged image $I_{merged}^c \in \mathbb{R}^{H' \times W' \times 3}$, where the concatenation is done on the width dimension if $W \leq H$ or on the height dimension if $H < W$, and $\max(H', W') = P$ for some hyperparameter $P \in \mathbb{Z}$. Then we use the following text prompt:

---

**Chain-of-Thought Prompt:**

---

*'You will act as a keyframe selection agent for a video reasoning task. During each inference, you will be given a grid image that contains multiple keyframes sampled from a long video. The keyframes are aligned from top to down or from left to right, following their temporal order. You will also be given a complex user query that implicitly or explicitly refers to one or more target objects in the video. You need to think in chain of thoughts to analyze each keyframes and find the best keyframes for each target object, where a segmentation model can find the target object in that frame with less effort. Your chain of thoughts should begin with what can be seen in each keyframe, how many objects in total fulfilling the requirements of the user query, etc. Some of the objects may be seriously obscured or blocked by other objects. Some of the objects may be camouflaged in their surroundings. Analyze each frame separately to get all the visible objects. This chain of thoughts should follow the output format:*

*"Chain of Thoughts:*

*- Frame 1: <analysis of frame 1>;*

*- Frame 2: <analysis of frame 2>;*

*...". For the analysis of each frame, you also have to follow the chain-of-thought format:*

*"- \*<question 1>\* <answer 1>;*

*- \*<question 2>\* <answer 2>;*

*...", where you have to ask question to yourself and answer it. Your answer should be as detailed as possible. You should start with broader questions, like "what can be seen in the frame?" to some detailed questions like "are there any other objects that haven't be listed" and "how many and which objects satisfy the user query?". There will be many questions and answers in the analysis of each frame, helping you to fully understand the frame. The actual questions vary by cases. Generate the in-depth questions and answers based on your previous analysis. Your thinking process aims to find the keyframe for each target object of interest (find all the target objects, each of which corresponds to a keyframe) related to the user query. Lastly, you have to output a list of dictionary with a format:*

*"Output list: [{object_index: 1, keyframe: k_1, object_description: <description of the object 1 in keyframe k_1>}, {object_index: 2, keyframe: k_2, object_description: <description of the object 2 in keyframe k_2>}, ...]"*

*, where each element in the list is a dictionary with three items, with object index, keyframe index, object description. k is the k-th keyframe in the grid image. object_index is a numbering integer starting with 1. object_description implies the description for that object in a particular frame, helping the model to find the object in that particular frame. For example, a valid element in an output list can be like 'Output list: [{object_index: 1, keyframe: 4, object_description: "the man at the top left corner of the image"}]'. You have to include all objects that fulfill the requirements in the user query. Include the objects even if it is only partially visible. While choosing the keyframe for any object, you should prioritize those frames where objects are not overlapped. This will help model to better recognize the object. Keep the output list in text format. Don't use json formatting. The output list begins with the prefix "Output list: ", followed by a square bracket with multiple curly brackets. The square bracket should be in the same line, following the format "Output list: [...]". Don't start with a new line.*

*Here is a grid image with {num_keyframes} keyframes. The user query is "{query}". Follow the instruction and output the index of the best keyframe.'*

---

where {*num_keyframes*} and {*query*} respectively refer to the number of keyframe candidates $T'$ and user query $q$ in Sec. 3.2. Lastly, we parse the output list to retrieve the keyframe selectivity $S$.

## APPENDIX B    COT-RVS (ONLINE) COT PROMPT DETAILS

In the online version of CoT-RVS, an incoming frame $I_t$ will be sent to the MLLM keyframe selector (*i.e.,* LLaVA-7B Liu et al. (2023a)) for binary selectivity when $t = n\xi + 1$. Below is the CoT prompt for our online CoT-RVS:

---
**Online Chain-of-Thought Prompt:**

---
*'Consider the query "{query}" for an object tracking task. First analyze what can be seen in the input image and then output a simple answer with Yes or No to justify whether the input image is suitable as a keyframe. A good keyframe is an image that contains the target object. Please follow the output format "The image contains .... Therefore, the justification of using this image as keyframe is <Yes./No.>"'*

---

where *{query}* represents the user query $q$. The output of the MLLM will be parsed for binary selectivity $S_t$.

## APPENDIX C    HARDWARE DETAILS

CoT-RVS with GPT-4o is implemented with OpenAI API and one RTX-4090 GPU. Other experiments (*i.e.,* the CoT-RVS-Gemma3-12B and the online CoT-RVS-LLaVA1.5-7B) are implemented with two RTX-4090 GPUs.

## APPENDIX D    T-REASONVOS DATASET

ReasonVOS contains 91 videos (18 from MeViS Ding et al. (2023a), 26 from MOSE Ding et al. (2023b), 37 from BURST  Athar et al. (2023), and 10 from VIPSeg Miao et al. (2022)) and 458 queries, which was proposed to effectively evaluate the performance of reasoning segmentation in videos Bai et al. (2024). While being widely used, this benchmark falls short in giving a fair evaluation on temporal reasoning ability. In particular, some considerable number of queries are temporally insensitive and thus less challenging, where the context only refers to the appearance or function of the relevant video objects unrelated to the actions or object movements. Thus, we believe a new dataset is both timely and essential in clearly demonstrating the efficacy of CoT-RVS on temporal reasoning. Specifically, we sampled 180 video-query pairs that correspond to 45 videos (8 from MeViS, 18 from MOSE, 16 from BURST, and 3 from VIPSeg) from ReasonVOS to produce a new benchmark called T-ReasonVOS. We remove time-insensitive queries with target objects visible from the beginning to the end of the video, such as *"The mode of transportation capable of transporting the largest group of people"*, where users can make decisions based on independent frames with image segmentation models. Surviving data in T-ReasonVOS after removing time-insensitive counterparts focus on videos where the target objects are visible only in a short interval, or queries specifying actions and object movements, *e.g., "The thing that has made a 180-degree turn."* This requires models to "think" in the temporal dimension to find the correct objects of interest. The T-ReasonVOS dataset and the selection code will be released.

## APPENDIX E    COT QUALITATIVE RESULTS

In this appendix, we will demonstrate the CoT outputs of the keyframe selector.

**CoT-RVS (Original).**    Refer to Appx. A for the input format. Below is two qualitative result with $T' = 7$ and $T' = 4$, respectively.

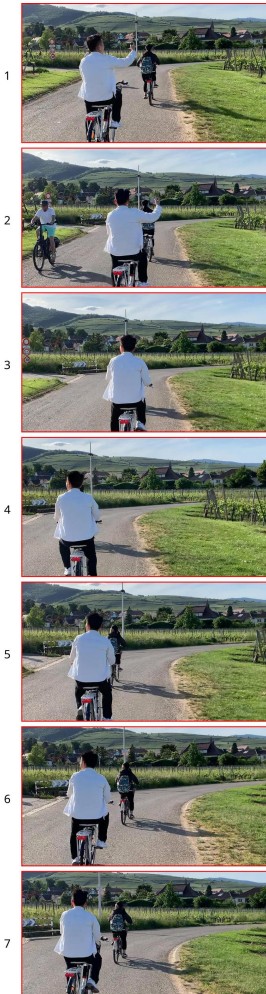

Figure A1: Input image $I_{merged}^c$.

**CoT-RVS Example 1 (GPT-4o):**

**Input query:** *Please segment all the visible subjects that are using some kind of tranportation tool.*

**Chain of Thoughts:** - *Frame 1:* - \*What can be seen in the frame?\* There are two individuals visible, both are using bicycles. - \*Are any subjects using transportation tools?\* Yes, both subjects are using bicycles. - \*How many and which objects satisfy the user query?\* There are two individuals, both satisfy the query as they are using bicycles. - \*Where is the first subject located?\* The first subject, wearing a white shirt, is located at the left center of the frame. - \*Where is the second subject located?\* The second subject, wearing a backpack, is further down the road at the center of the frame.

- *Frame 2:* - \*What can be seen in the frame?\* Three individuals are visible. Two in the foreground and one in the background, all using bicycles. - \*Are any subjects using transportation tools?\* Yes, all three subjects are using bicycles. - \*How many and which objects satisfy the user query?\* Three individuals satisfy the query using bicycles. - \*Where is the first subject located?\* The first subject, wearing a white shirt, is at the center left side of the frame. - \*Where is the second subject located?\* The second subject, in a light-colored shirt, is to the left of the first subject, occupying the left side of the frame. - \*Where is the third subject located?\* The third subject, with a backpack, is in the center back of the frame.

- *Frame 3:* - \*What can be seen in the frame?\* Two individuals are visible; both are on bicycles. - \*Are any subjects using transportation tools?\* Yes, both are using bicycles. - \*How many and which objects satisfy the user query?\* There are two individuals using bicycles satisfying the query. - \*Where is the first subject located?\* The first subject, wearing a white shirt, is centrally located in the frame. - \*Where is the second subject located?\* The second subject, wearing a backpack, is near the center-back of the frame.

- *Frame 4:* - \*What can be seen in the frame?\* Two individuals are visible, both using bicycles. - \*Are any subjects using transportation tools?\* Yes, both individuals are using bicycles as transportation. - \*How many and which objects satisfy the user query?\* Two subjects satisfy as they ride bicycles. - \*Where is the first subject located?\* The first subject, in a white shirt, is centrally situated. - \*Where is the second subject located?\* The second subject, with a backpack, is at the edge of the frame near the center back.

- *Frame 5:* - \*What can be seen in the frame?\* Two individuals are seen on bicycles. - \*Are any subjects using transportation tools?\* Yes, both are on bicycles. - \*How many and which objects satisfy the user query?\* Two subjects, both riding bicycles, satisfy this query. - \*Where is the first subject located?\* The first subject, wearing a white shirt, occupies the central area of the frame. - \*Where is the second subject located?\* The subject with a backpack is in the center background of the image.

- *Frame 6:* - \*What can be seen in the frame?\* Two individuals on bicycles. - \*Are any subjects using transportation tools?\* Yes, both are using bicycles. - \*How many and which objects satisfy the user query?\* Two individuals satisfy the query as they use bicycles. - \*Where is the first subject located?\* The first subject, in a white shirt, is placed centrally. - \*Where is the second subject located?\* The subject with a backpack appears in the central background.

- *Frame 7:* - \*What can be seen in the frame?\* Two individuals on bicycles. - \*Are any subjects using transportation tools?\* Yes, both individuals are using bicycles. - \*How many and which objects satisfy the user query?\* Two individuals meet the criteria, both on bicycles. - \*Where is the first subject located?\* The first subject, in a white shirt, is in the center. - \*Where is the second subject located?\* The individual with a backpack is further down, towards the center.

***Output list:*** [{object_index: 1, keyframe: 1, object_description: "the individual in a white shirt riding a bicycle at the left center of the frame"}, {object_index: 2, keyframe: 1, object_description: "the individual with a backpack on a bicycle at the center further down the road"}, {object_index: 3, keyframe: 2, object_description: "the individual in a light-colored shirt on a bicycle at the left side of the frame"}]

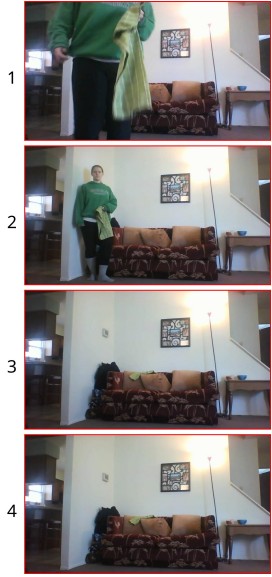

Figure A2: Input image $I^c_{merged}$.

**CoT-RVS Example 2 (GPT-4o):**

**Input query:** *Which item in the scene serves the purpose of drying hands after they've been washed?*
**Chain of Thoughts:** - *Frame 1:* - \*What can be seen in the frame?\* A person is holding a green towel in hand, standing near a sofa. - \*Is there any object that serves the purpose of drying hands?\* Yes, the green towel being held by the person can serve that purpose. - \*Are there any other objects that haven't been listed?\* No, the towel is the primary object fulfilling the query in this frame.
- *Frame 2:* - \*What can be seen in the frame?\* The person is standing next to a sofa, holding a green towel in both hands. - \*Is the towel clearly visible?\* Yes, the towel is clearly visible and being held prominently. - \*Does the towel satisfy the user query?\* Yes, it serves the purpose of drying hands after washing.
- *Frame 3:* - \*What can be seen in the frame?\* The person is no longer visible. The towel is now laying on the back of the sofa. - \*Does any object fulfill the purpose of drying hands?\* Yes, the towel on the sofa still fulfills this purpose. - \*Is the towel more visible in this frame than previous frames?\* The towel is distinct but lay still; it might not be as instantly recognizable as when held by the person in previous frames.
- *Frame 4:* - \*What can be seen in the frame?\* The frame is similar to Frame 3; the towel is on the back of the sofa, and no people are present. - \*Does it fulfill the purpose of drying hands?\* Yes, the towel on the sofa does. - \*Is there any difference from Frame 3 regarding the towel's visibility?\* No significant differences are noted regarding visibility.
*Output list:* [{object_index: 1, keyframe: 2, object_description: "the green towel held prominently by the person"}]

**CoT-RVS (Online).** Refer to Appx. B for the input format. Below is the qualitative results of two sampled frames $I_0$ and $I_{56}$ from the same video, where $\xi$ is set to 4:

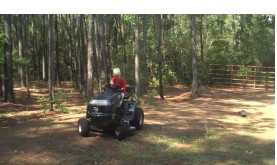

Figure A3: Input frame $I_0$.

**CoT-RVS Online Example 1 (LLaVA1.5-7B):**

**Input query:** *When parents see their children in potential danger, they are often concerned and ready to help. Who is the parent in the scene?*
**Chain of Thoughts:** *The image contains a person riding a four-wheeler in a wooded area. There is no clear indication of a parent in the scene. Therefore, the justification of using this image as a keyframe is No.*

Figure A4: Input frame $I_{56}$.

**CoT-RVS Online Example 2 (LLaVA1.5-7B):**

**Input query:** *When parents see their children in potential danger, they are often concerned and ready to help. Who is the parent in the scene?*
**Chain of Thoughts:** *The image shows the man and the child in a field, with a fence in the background. The man appears to be concerned about the child's safety while they are on the motorcycle. Therefore, the justification of using this image as a keyframe is Yes.*

## APPENDIX F    EFFICIENCY AND COST ANALYSIS

While efficiency is not our main focus, as CoT's complex reasoning usually requires higher computational complexity, we include the efficiency and cost analysis with each CoT-RVS component and the comparison with VideoLISA for your reference. All latency and memory measurements are based on 324-sized frames and 8 keyframe candidates for the MLLM selector. The API cost only applies to GPT-4o, which is a closed-source model. Results are reported on RTX4090.

In specific, Tab. A1 includes the latency, local memory footprint, and API cost of each module used in CoT-RVS and its online version. The end-to-end analysis of CoT-RVS is further presented in Tab. A2.

For the online CoT-RVS, the per-frame latency within a $\xi$-frame sampling window is approximated by $\frac{\alpha+S\beta+(\xi-1)\gamma}{\xi}$, where $\alpha$ is the MLLM inference latency, $\beta$ is the segmentation agent latency, $\gamma$ is the tracking latency per frame, and $S$ is the binary indicator for frame selection. This yields an

upper bound of $\frac{\alpha + S\beta}{\xi} + \gamma$ for average latency, where $\gamma$ is relatively small (0.04 in Tab. A1) and $\xi$ is a hyper-parameter. Please refer to Tab. A3 for actual latency measurements across different MLLMs.

Table A1: Latency of each CoT-RVS component

| Module | Latency(s) | Local Memory footprint (GB) | API cost (US$) |
|---|---|---|---|
| LLaVA1.5-7B | 1.71 | 15.6 | - |
| Gemma3-12B | 63.95 | 24.8 | - |
| GPT-4o | 12.01 | - | 0.012 |
| Seg-Zero | 4.57 | 18.1 | - |
| SAM2 | 0.04† | 2.6 | - |

† Average per-frame tracking latency.

Table A2: End-to-end cost comparison with VideoLISA.

| Method | Latency(s) | Local Memory footprint (GB) | API cost (US$) |
|---|---|---|---|
| VideoLISA (OTSA) | 3.46 | 11.7 | - |
| CoT-RVS-Gemma3-12B | 69.79 | 24.8 | - |
| CoT-RVS-GPT-4o | 17.85 | 18.1 | 0.012 |

Table A3: Ablation study for online keyframe selector on Refer-DAVIS-17. * varies depending on the OpenAI server busyness. † is implemented on an RTX4090.

| Model | $\mathcal{J}\&\mathcal{F}$ | Avg. Time (sec.) |
|---|---|---|
| Qwen-2.5-VL-3B Bai et al. (2025) | 72.5 | 0.18† |
| LLaVA1.5-7B Liu et al. (2023a) | 73.9 | 0.24† |
| GPT-4o Hurst et al. (2024) | 73.2 | 0.27* |

# APPENDIX G   ROBUSTNESS ANALYSIS

The quality of keyframe selection is influenced by the choice of MLLM. To evaluate the adherence of MLLM to our CoT prompt in offline setting, we compute the percentage of outputs where the MLLM produces at least one valid object instance per expression (termed "valid outputs"). Results in Tab. A4 show that larger, closed-source MLLMs such as GPT-4o offer significantly more robust and reliable reasoning, while smaller models like LLaVA1.5-7B and Qwen2.5-VL-3B fail to follow the output format as described in the CoT prompt (refer to Appx. A), thereby resulting in errors of data parsing. Prior works often underutilize this potential of large closed-source models in video segmentation tasks.

Table A4: Robustness analysis of CoT-RVS (original) with different keyframe selector models.

| Dataset | MLLM | #Valid outputs/#Expressions | Percentage | $\mathcal{J}\&\mathcal{F}$ | $\mathcal{J}$ | $\mathcal{F}$ |
|---|---|---|---|---|---|---|
| MeViS (val_u) | LLaVA1.5-7B | 0/785 | 0%‡ | - | - | - |
| | Qwen2.5-VL-3B | 0/785 | 0%‡ | - | - | - |
| | Gemma3-12B | 715/785 | 91.1% | 52.6 | 48.2 | 57.0 |
| | GPT-4o | 761/785 | 96.8% | **60.1** | **56.0** | **64.2** |
| ReasonVOS | LLaVA1.5-7B | 0/785 | 0%‡ | - | - | - |
| | Qwen2.5-VL-3B | 0/785 | 0%‡ | - | - | - |
| | Gemma3-12B | 273/458 | 59.6% | 50.7 | 47.5 | 54.0 |
| | GPT-4o | 444/458 | 96.9% | **65.5** | **62.4** | **68.7** |

‡ Incorrect output format.

# APPENDIX H   MODULARITY ANALYSIS

We have demonstrated in main paper that GPT-4o and Gemma3 can be used as MLLM keyframe selector. In this appendix, we further validate the modularity of CoT-RVS, where we adopt LISA Lai et al. (2023) and Cutie Cheng et al. (2024) alternatives to $F_{seg}$ and $F_{vid}$, respectively. We conduct this ablation under the Refer-DAVIS-2017 benchmark with Gemma3-12B as keyframe selector. Tab. A5

shows that both Seg-Zero and SAM2 in our default CoT-RVS can be substituted without significantly sacrificing the performance.

Table A5: Ablation study on Ref-DAVIS-17 using CoT-RVS-Gemma3-12B with different $F_{seg}$ and $F_{vid}$.

| $F_{seg}$ | $F_{vid}$ | $\mathcal{J\&F}$ | $\mathcal{J}$ | $\mathcal{F}$ |
|---|---|---|---|---|
| LISA-13B-Llama2 | Cutie | 68.8 | 65.0 | 72.5 |
| LISA-13B-Llama2 | SAM2 | 71.1 | 67.3 | 75.0 |
| Seg-Zero-7B | Cutie | 72.1 | 68.5 | 75.7 |
| Seg-Zero-7B | SAM2 | **74.6** | **70.9** | **78.3** |

## APPENDIX I  FAILURE CASE

We demonstrate a failure case of CoT-RVS in Fig. A5. Specifically, our empirical experience shows that: when multiple object instances are visible in the video, the GPT-4o fails to identify the individual instance. Instead, similar objects are packaged within an element of the output list, *e.g., "the six geese near the center of the image"*. This results in the further failure of the reasoning image segmentation model. Furthermore, it is difficult to separate different instances with textual descriptions, which will be sent to the image segmentation models for each instance. Eventually, only one instance is presented in the output mask sequence in this case.

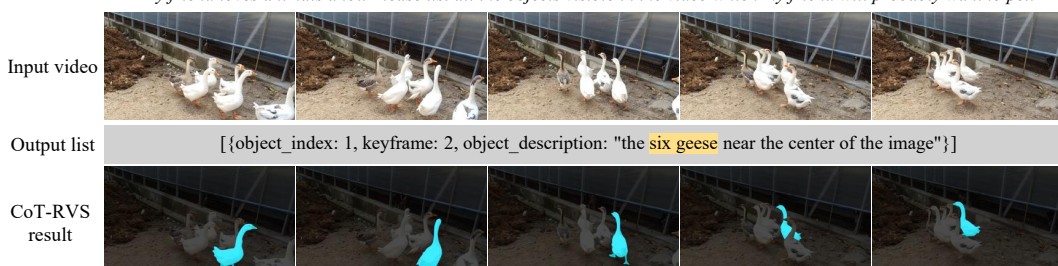

Figure A5: **A failure case of CoT-RVS.**

## APPENDIX J  DISCUSSION

**Keyframe selector choice.** We adopt GPT-4o as the keyframe selector in the original CoT-RVS, while we also conduct experiments on additional open-sourced models for better reproducibility. We notice that smaller models, like LLaVA1.5-13B Liu et al. (2023a) or Qwen2.5-VL-3B Bai et al. (2025), failed to follow our detailed CoT prompts. Gemma3-12B produces reasonable results and performs well in the Referring VOS datasets, while falling short in difficult reasoning tasks. More comprehensive experiments of open-sourced models on larger models, such as Llama4, remain unexplored due to the limited computational resources.

**Dataset quality.** We demonstrate an example from the ReasonVOS benchmark Bai et al. (2024) of CoT-RVS in Fig. A6. Specifically, we notice that CoT-RVS segments the hoodie when asked for an object to alleviate the cold, while the ground-truth mask sequence refers to a blanket. This showcases that some of the reasoning queries are too ambiguous even for humans. While the benchmark is not our major focus and contribution, we have attempted to avoid such ambiguous queries when sampling the T-ReasonVOS dataset.

**Future directions.** In this work, we are excited to have made significant strides in addressing the challenges of Reasoning Video Object Segmentation (Reasoning VOS) and pushing the boundaries further into Reasoning Video Instance Segmentation (Reasoning VIS) and Online Reasoning VOS. The qualitative and quantitative results of CoT-RVS are indeed impressive, showcasing its potential

*"After sitting on the sofa and watching TV for a while, I started to feel a bit cold. Which object is most likely to help me alleviate the cold?"*

Figure A6: **Ambiguous case.** Zoom in for details.

to enhance segmentation tasks. However, we acknowledge that our evaluation methods for Online Reasoning VIS and Reasoning VOS were not as rigorous as desired, where we only presented the qualitative results instead of quantitative results, primarily due to the limitations of available benchmarks in these tasks. This presents an opportunity for future research, where we aim to develop more comprehensive evaluation frameworks and datasets to thoroughly assess and refine the capabilities of CoT-RVS, ensuring it continues to advance the field of video segmentation.

**Social impacts.** Potential misuse of the CoT-RVS application could include its use in unauthorized surveillance or the creation of deceptive media. For example, individuals or organizations could leverage the technology to track people without their consent, leading to privacy violations and potential harassment. We believe that licensing can be a useful tool to prevent such issues, where we will establish specific terms and conditions governing the use of the application by the time we release the code.

## APPENDIX K  MORE COMPARISON

Fig. A7 contains three examples from T-ReasonVOS. We show that our proposed CoT-RVS consistently outperforms the state-of-the-art VideoLISA Bai et al. (2024) on temporally sensitive queries.

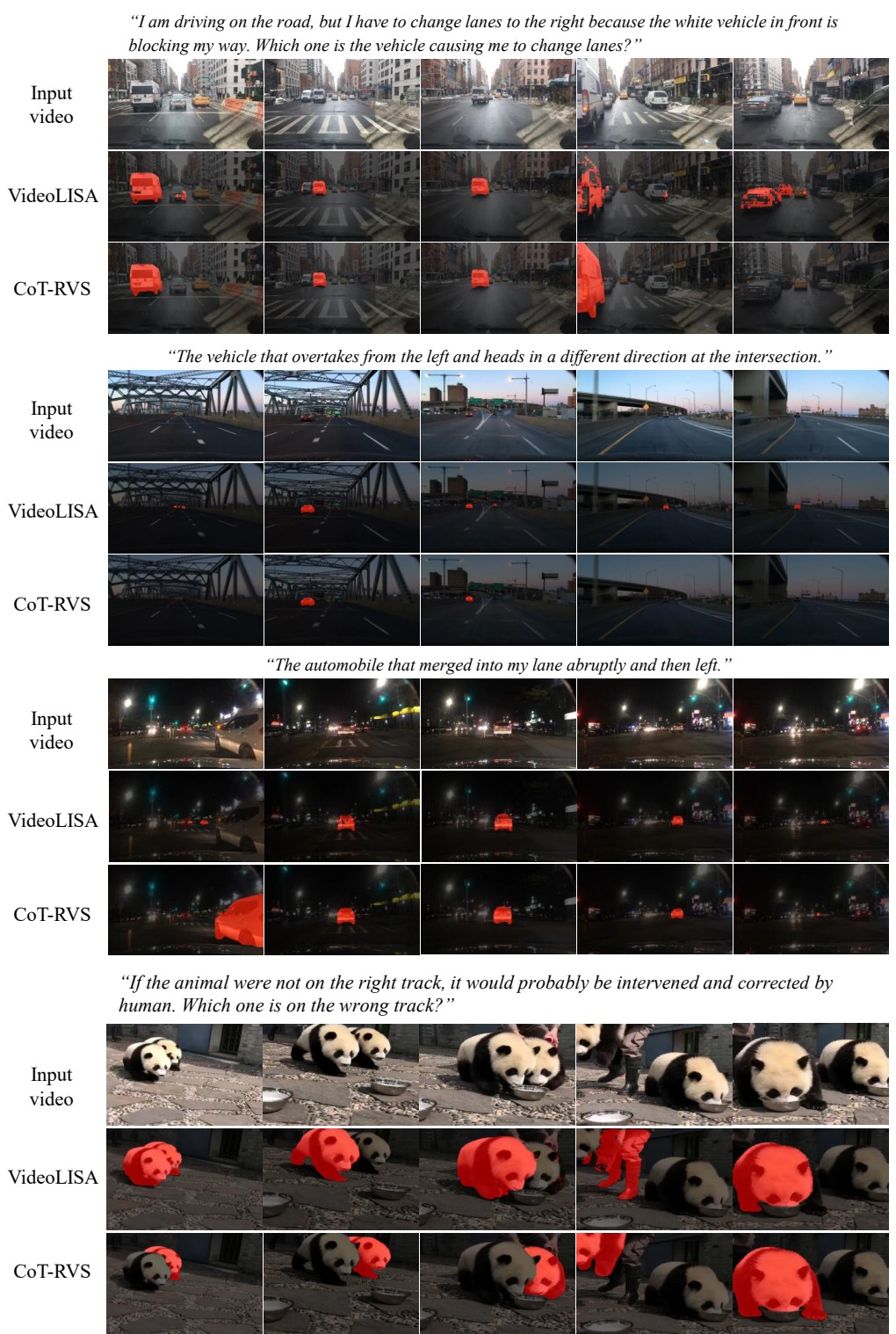

Figure A7: **More qualitative comparison.** Refer to the anonymous website in the supplementary materials for video demo.

