# OpenReview forum: "CoT-RVS: Zero-Shot Chain-of-Thought Reasoning Segmentation for Videos"
_ICLR.cc/2026/Conference — ICLR 2026 Poster_

### Official Review · Reviewer_EGe3 · 2025-10-30

**Soundness:** 3
**Presentation:** 3
**Contribution:** 3
**Rating:** 6
**Confidence:** 4

**Summary:**

The paper introduces CoT-RVS, a zero-shot, and training-free framework for Reasoning Video Object Segmentation (Reasoning VOS). The primary goal is to segment objects in a video based on complex, implicit, or temporally-sensitive text queries.The core idea is to decouple high-level temporal-semantic reasoning from low-level segmentation and tracking. The method uses a modular, multi-agent pipeline:MLLM Keyframe Selector, Reasoning Image Segmenter, Video Processor.

**Strengths:**

1. A key aspect of the paper's design is its problem decomposition. Instead of using a single model to handle both temporal reasoning and pixel segmentation, it decouples the task into two stages: 1) First, an MLLM ($F_{key}$) performs high-level temporal-semantic reasoning to identify the most suitable keyframe and target description ; 2) Subsequently, specialized vision models ($F_{seg}$, $F_{vid}$) execute segmentation and tracking based on the clear instructions provided by the MLLM . This architecture shifts the challenge of complex temporal understanding from the pixel domain to the semantic reasoning domain.
2. The paper provides sufficient experiments to validate its method. It was tested not only on four standard benchmarks (MeViS, Refer-DAVIS-17, ReVOS, ReasonVOS) but also on a specially constructed, temporally-sensitive T-ReasonVOS dataset to validate its core hypothesis. The experimental results show that the method achieves performance exceeding the compared SOTA methods on these benchmarks, particularly on T-ReasonVOS
3. The framework is designed to be training-free and modular. This design allows for the replacement of the MLLM component (e.g., GPT-4o, Gemma3) without retraining. The architecture also supports an extension from single-object (VOS) to multi-instance (VIS) segmentation (via the MLLM outputting a list) and is refactored for an online version (via periodic keyframe updates ).

**Weaknesses:**

1. the paper's most compelling conclusion—superiority on temporally-sensitive tasks—relies heavily on the newly created T-ReasonVOS dataset . Given that this dataset was manually filtered by the authors and is not yet publicly released, this raises concerns regarding reproducibility and potential selection bias.

2. the ablation studies are incomplete. While they validate the importance of the CoT process and sampling rate, the paper fails to fully explore its claimed modularity by, for instance, swapping the $F_{seg}$ (Seg-Zero) or $F_{vid}$ (SAM2) components to test the framework's generalizability

3. the system is highly dependent on the MLLM's adherence to specific prompt formats.

4. The framework's practical viability is questionable due to high computational latency and cost.

**Questions:**

See the weaknesses.

---

> ### Author Response · Authors · 2025-11-23
> **Author Individual Response to Reviewer EGe3**
>
> ## W1 T-ReasonVOS reproducibility.
> As specified in the reproducibility statement, we will release T-ReasonVOS **upon acceptance** to maintain anonymity. You may find some examples from T-ReasonVOS in our supplementary material for your reference, such as the ones with query *"The vehicle that overtakes from the left and heads in a different direction at the intersection."* and "*If the animal were not on the right track, it would probably be intervened and corrected by human. Which one is on the wrong track?*". Besides, we would like to mention that CoT-RVS also outperforms state-of-the-art in existing benchmarks. The T-ReasonVOS is used to verify the main hypothesis: better temporal-semantic reasoning improves video segmentation quality.
>
> ## W2 Swap $F_{seg}$ and $F_{vid}$.
> We examine the alternatives for $F_{seg}$ and $F_{vid}$ of CoT-RVS-Gemma3-12B, using LISA and Cutie [1], on Ref-DAVIS-17 benchmark in the following table:
>
> | $F_{seg}$            | $F_{vid}$  | J&F  | J    | F    |
> |:-----------------|:-------|------|------|------|
> | LISA-13B-Llama2  | Cutie  | 68.8 | 65.0 | 72.5 |
> | LISA-13B-Llama2  | SAM2   | 71.1 | 67.3 | 75.0 |
> | Seg-Zero-7B      | Cutie  | 72.1 | 68.5 | 75.7 |
> | Seg-Zero-7B      | SAM2   | **74.6** | **70.9** | **78.3** |
>
> Results show both Seg-Zero and SAM2 in our default version of CoT-RVS can be substituted without significantly deteriorating the performance.
>
> [1] Cheng et al. "Putting the object back into video object segmentation.", CVPR 2024.
>
> ## W3 Specific prompt format.
> While we agree that our framework is subject to MLLM's adherence to our specified prompt format, we would like to clarify that our CoT-RVS is quite prompt-invariant against different MLLMs, where Gemma3 and GPT-4o can be applied for the original CoT-RVS, and Qwen-2.5-VL, LLaVA1.5, and GPT-4o are studied for the online CoT-RVS. To analyze the MLLM's adherence to our CoT prompt, we also conduct robustness ablation study in Table A4 in Appx. G for different MLLMs, where success rate of output object list is reported. In our opinion, this should be regarded as input setup rather than limitation, which is indeed common in highly-cited zero-shot MLLM's applications, *e.g.* [2] generates bounding boxes with the prompt *"You are an intelligent bounding box generator. I will provide you with ..."* and [3] creates robot planning with the prompt *"You are highly skilled in robotic task planning, breaking down intricate and long-term tasks into distinct primitive actions ..."*
>
> [2] Lian et al. "LLM-grounded Diffusion: Enhancing Prompt Understanding of Text-to-Image Diffusion Models with Large Language Models", TMLR 2024
>
> [3] Hu et al. "Look Before You Leap: Unveiling the Power of GPT-4V in Robotic Vision-Language Planning", ICRA 2024
>
> ## W4 Computational latency and cost.
> Latency and cost analyses are provided in Appx. F, where we consider this cost acceptable. Specifically, for the offline mode, CoT-RVS-GPT-4o takes around 19 seconds for a 60-frame long video. For the online extension, CoT-RVS-LLaVA-7B achieves approximately 4.2 FPS on a single RTX4090.

---

### Official Review · Reviewer_bGVR · 2025-10-31

**Soundness:** 3
**Presentation:** 3
**Contribution:** 3
**Rating:** 6
**Confidence:** 4

**Summary:**

This paper introduces CoT-RVS, a training-free framework for Reasoning Video Object Segmentation. Instead of end-to-end finetuning, the framework is modular. It decomposes the R-VOS task into three distinct stages: an MLLM-based Keyframe Selector that uses zero-shot Chain-of-Thought prompting to perform temporal-semantic reasoning, an off-the-shelf Reasoning Image Segmentation Model to generate a key mask, and a Video Processor (i.e., SAM2) to track the object throughout the video. The paper demonstrates that this framework achieves state-of-the-art performance on several R-VOS benchmarks, with extensions to Reasoning Video Instance Segmentation and online streaming video.

**Strengths:**

1. The core idea of a training-free, modular framework is compelling. This approach cleverly composes the strengths of large pre-trained models to bypass the need for task-specific finetuning.
2. The quantitative results are strong, showing the framework outperforms prior SOTA methods on multiple benchmarks.
3. The framework is shown to be flexible. The authors demonstrate its applicability beyond standard R-VOS, with extensions for Instance Segmentation and Online Reasoning VOS.

**Weaknesses:**

1. The primary weakness is that the framework is an integration of existing, powerful components rather than a new technical method. It "stitches together" a large MLLM, an image segmenter, and a video tracker. The system-level design is a valid engineering contribution, but the paper would be stronger if it more clearly articulated this as a novel contribution beyond "stitching".
2. The paper's central claim, embedded in its title, is the power of the Chain-of-Thought reasoning process. However, this claim is supported almost exclusively by a qualitative ablation in Figure 8. The paper lacks a quantitative ablation study regarding CoT.

**Questions:**

1. What is the key difference between this work and ThinkFast?
2. According to Table A4 in Appendix G, LLaVA-1.5-7B and Qwen2.5-VL-3B are poor at key frames selection. How are the results achieved in the main paper?
3. Why only CoT-RVS-LLaVA supports online mode, while Gemma and GPT-4o do not support?

---

> ### Author Response · Authors · 2025-11-23
> **Author Individual Response to Reviewer bGVR**
>
> ## W1 Novelty on model integration.
> While the reviewer perceive our CoT-RVS as integration of existing powerful modules, possibly due to its training-free nature, we respectfully disagree with this conclusion. Our approach does not involve training or fine-tuning any models, and we believe this design choice should be viewed as a strength rather than a weakness, as it enables broad applicability, ease of deployment, and reproducibility without additional computational overhead.
>
> In the realm of Computer Vision and Robotics, several recent training-free approaches demonstrate this trend as well:
>
> [1] From Images to Textual Prompts: Zero-shot Visual Question Answering with Frozen Large Language Models, CVPR 2023
>
> [2] LLM-grounded Diffusion: Enhancing Prompt Understanding of Text-to-Image Diffusion Models with Large Language Models, TMLR 2024
>
> [3] Look Before You Leap: Unveiling the Power of GPT-4V in Robotic Vision-Language Planning, ICRA 2024
>
> We respectfully clarify that our contributions lie in the following:
>
> 1. A modular, zero-shot reasoning VOS framework that coordinates MLLMs, image segmenters, and video processors in both offline and online settings, without any training or fine-tuning, unlike prior methods.
> 2. A keyframe selection pipeline based on CoT prompting, specifically tailored for temporal reasoning, where MLLMs are required to both localize and describe scene-relevant frames, beyond simple object retrieval.
> 3. An online reasoning extension that adaptively re-selects keyframes at inference time, a feature rarely explored in existing R-VOS systems, enabling streaming video processing.
>
> This design enables broad model generalization (e.g., GPT-4o, Gemma, LLaVA), which we believe reflects a non-trivial engineering and algorithmic contribution. We will further clarify these distinctions and contributions explicitly in the final revision.
>
> ## W2 Quantitative evaluation of CoT.
> Our quantitative evaluation focuses on the final mask sequences, mainly because of two reasons: First, there is no absolute ground-truth for the reasoning steps. Second, there is no absolute keyframe for any target instance, where the MLLM’s output is considered successful as long as the target instance can be viewed clearly. Since we do not fine-tune the MLLM, we consider comparing ground-truth CoT unnecessary and somewhat nonsensical. Notwithstanding, as an ablation study, we suggest the reviewer to  refer to Table 6 for hyper-parameter $\xi$, where the CoT length of MLLM is in inverse proportion to $\xi$. Specficially, we found a significant performance drop when applying long CoT, possibly due to the long context and hallucination.
>
> ## Q1 Difference with "ThinkFast".
> We would like to clarify the question whether the reviewer is referring to "ThinkFirst". If so, we would highlight that ThinkFirst is an image-based segmentation system without temporal understanding, which often results in serious temporal inconsistency as demonstrated in Figure 6 and Table 5.
>
> ## Q2 Table A4 results.
> We mentioned in the caption that Table A4 shows the success rate of keyframe selection for original CoT-RVS, which focuses on **offline** video segmentation. In contrast, LLaVA1.5’s experiments in Table 1-4 of the main paper are from the **online** extension of CoT-RVS. We will further clarify this in our revised manuscript. We attribute this performance gap to the fundamental difference between the pipeline designs and the distinct prompts of original CoT-RVS and its online extension, where smaller models like LLaVA1.5 may fail to incorporate complex temporal-semantic reasoning given the long CoT prompt for original CoT-RVS.
>
> ## Q3 Online CoT-RVS with Gemma and GPT-4o.
> There is **no restriction** on the MLLM choice in online mode. We analyzed different MLLMs in online CoT-RVS in Table A3 in Appx. F. Table A3 shows that LLaVA1.5-7B achieves comparable results with GPT-4o. Considering the inference cost, we adopt LLaVA1.5 as our keyframe selector in online CoT-RVS. We will provide reference to Appx. F in the revised main paper to avoid confusion.

---

### Official Review · Reviewer_7g6P · 2025-10-31

**Soundness:** 3
**Presentation:** 3
**Contribution:** 2
**Rating:** 4
**Confidence:** 3

**Summary:**

This paper proposed a novel training-free framework called CoT-RVS. This method leverages the zero-shot CoT capability of MLLM to select the optimal keyframes for segmentation. This framework outperforms existing methods both qualitatively and quantitatively.

**Strengths:**

1.	The proposed CoT-RVS architecture can be seamlessly extended to online video stream processing (Online Reasoning VOS), showing strong practical potential.
2.	The method primarily relies on the zero-shot capability of MLLMs, requiring no additional training and exhibiting good generalization ability.

**Weaknesses:**

1.	From a pipeline perspective, the proposed method shows limited distinction from prior training-free works such as AL-Ref-SAM. The explicit introduction of Chain-of-Thought reasoning has also been explored in previous reasoning segmentation studies, indicating limited methodological novelty.
2.	Although the introduction emphasizes the importance of reasoning over temporal-semantic correlation, this motivation is not clearly reflected in either the method design or the experiments.
3.	The MLLM keyframe selector and reasoning image segmentation model communicate through text, which may lead to inconsistencies between the described instance and the segmented target when multiple similar objects exist.
4.	The method requires long-chain reasoning on each sampled frame, limiting the length and complexity of the input video, and potentially causing issues such as forgetting or hallucination. This restricts scalability to complex or long-duration videos.
5.	Using GPT-4o as the base model makes the comparison with other methods unfair; parameter amount, inference cost, and latency introduced by the CoT mechanism should be reported.
6.	The performance on Reasoning VOS tasks using LLaVA-1.5 and Gemma-3 is not significant, as shown in Table 4.

**Questions:**

See weaknesses.

---

> ### Author Response · Authors · 2025-11-23
> **Author Individual Response to Reviewer 7g6P**
>
> ## W1 Comparison with AL-Ref-SAM2 and CoT works.
>
> First, AL-Ref-SAM2 was proposed to solve referring VOS, while our work tackles much more challenging cases like implicit queries and temporally sensitive queries. Our empirical results also show that the performances of prior work severely drop when given challenging text queries. Second, our paper is a pioneering work to apply chain-of-thought reasoning in VOS, where we have shown in ablation studies how it improves the performance. Third, even though both CoT-RVS and AL-Ref-SAM2 adopt GPT-4 and SAM2, we outperform AL-Ref-SAM2 by approximately 5% in referring VOS benchmark like Ref-DAVIS-17, showcasing the importance of CoT reasoning in temporal dimension. Lastly, our CoT-RVS can be adapted to online reasoning VOS, while AL-Ref-SAM2 supports only offline videos.
>
> Regarding prior CoT works, we would like to highlight the novelty of applying CoT for temporal-semantic relationship. As a pioneering work in VOS, it is beyond straightforward to explore how to improve reasoning VOS performance with close-source MLLM, including the input/output formats, reasoning steps, and interaction with other modules.
>
> ## W2 Method design for temporal-semantic reasoning.
> Temporal-semantic relationship is analyzed in our MLLM keyframe selector, which depicts the per-frame semantics and conclude the target object(s) for keyframe selection in temporal dimension. Please refer to our MLLM’s prompt in Appx. A and sample CoT in Appx. E for more details.
>
> ## W3 Inconsistency between described instance and the segmented target.
> We agree that text-based communication may contribute to such inconsistency, which we also show one example in  Figure A5. However, text-based communication also brings additional advantages. For example, bounding-box based communication may perform poorly when encountering occlusions or slender objects.
>
> ## W4 Long video processing.
> Long video understanding and hallucination are both challenging problems in existing MLLMs. While long video processing is not our main focus, our designed framework allows user to choose a hyper-parameter as the number of keyframe candidates, where more keyframe candidates result in longer MLLM output with potential hallucination, and few keyframe candidates result in loss of video contents. Therefore, we show in Table 6 that a favorable range of keyframe candidates is between 4 and 8, where 16 results in significant performance drop. Refer to the supplemental videos for qualitative evaluation of long video object segmentation, where we have demonstrated videos with 235 frames (8 seconds), *e.g.* the examples with query *"the vehicle that overtakes from the left and heads in a different direction at the intersection."* and *"the automobile that merged into my lane abruptly and then left."*
>
> ## W5 GPT-4o's fairness and analysis.
> As specified in the introduction section, our goal is to enhance stronger reasoning ability through enabling diversely pre-trained MLLMs in Reasoning VOS. While we agree that GPT-4o is the strongest among evaluated models, none of the existing works have fully explored the mechanism to interact with close-source models like GPT-4o, thereby limited to inferior reasoning capability. In contrast, our CoT-RVS is the first work to be compatible with both close-source and open-source MLLMs. While maintaining training-free, it achieves comparable results with baselines using reasoning model in similar size, *i.e.* LLaVA1.5 and Gemma3. To further analyze the GPT-4o-based framework, we have provided GPT-4o's latency and inference cost in Table A1 in Appx. F.
>
> ## W6 LLaVA1.5 and Gemma3 performance.
> As per discussed in the limitation (Line 471), the CoT-RVS performance is bounded by the general reasoning capabilities of these LLMs. As shown in Table A4 in Appendix G, Gemma3 falls short in generating robust outputs when given implicit queries. Even so, we show that the the default backbone GPT-4o presents significant improvements, especially in difficult cases like T-ReasonVOS, as long as it generates robust outputs. We are encouraged by these results, which bridges LLM's potentials of scaling law and test-time compute to the segmentation task.  Notably, CoT-RVS-LLaVA1.5 outperforms other online methods in Table 3 and 4.

---

### Official Review · Reviewer_bVYo · 2025-11-01

**Soundness:** 3
**Presentation:** 3
**Contribution:** 3
**Rating:** 6
**Confidence:** 3

**Summary:**

This paper proposes a training-free referring video object segmentation by chain-of-thought capability of multimodal LLM in multi-agent framework, termed as CoT-RVS. CoT-RVS contains a multimodal LLM keyframe selector, a reasoning image segmentation model, and a video processor that track the masked selected object instances over the entire video. The CoT-RVS can be applied to offline video settings as well as online video settings. Experimental results show CoT-RVS outperforms prior methods on reasoning VOS benchmarks.

**Strengths:**

1) This paper proposes a framework to use the chain-of-thought capabilitity of pretrained MLLMs with prompting to perform reasoning first and segmentation then. Experiments show this design outperforms prior approaches and maintain training-free, being versatile with proprietary models.

2) The author also proposes a simple adaptation of this framework to online causal video settings, which is relatively underexplored by previous reasoning VOS methods but with great potential values.

**Weaknesses:**

There are some major concerns as listed follows:

1) For online video settings, could the authors please report the inference FPS and latency of this pipeline? It seems requiring many components' working to finish segmentation and might take too much time when handling online videos.

2) I was wondering whether this is pipeline agentic enough to be called an agent framework? The steps in this pipeline seem highly fixed and rule-based and lacking being autonomous enough. I would like to hear some author's discussion on this.

3) To what extent the performance gain is credited to the SAM2 as a video processor? As one knows, SAM2 excels at video object segementation given prompts. Is this pipeline still effective (and outperforming prior methods) when using SAM1 as a segmentation model?

**Questions:**

See weaknesses.

---

> ### Author Response · Authors · 2025-11-23
> **Author Individual Response to Reviewer bVYo**
>
> ## W1 Online FPS/latency.
>
> Refer to Appx. F for the efficiency and cost analysis, where time cost of online CoT-RVS is analyzed in Line 1077-1082. FPS can be derived from the average per-frame latency shown in Table A3 (i.e., FPS=1/avg. time)
>
> ## W2 Definition of agentic framework.
>
> Our pipeline can be autonomously inferenced with a simple python-based file, where we connect the user input with OpenAI API and parse the GPT’s output. In contemporary AI, an agentic framework is the system that specifies components and protocols for agent communication, coordination, reasoning, and decision-making. Recently, the advance of Multimodal Large Language Models (MLLMs) has led to a host of works that utilize them as a reasoning backbone in Agentic Systems [1]. According to [2], these MLLM-based agentic frameworks typically integrate modules for Memory, Structured Planning, tool/API use, and have shown excellent promise ranging from embodied simulators to video games. Closer to our work in visual perception, [3] introduces a segmentation agent to cut regions in VOS, [4] proposes an agentic framework composed of LLM, Grounding-DINO, and SAM2 for Referring Audio-Visual Segmentation (Ref-AVS). All these works refer their system as an agentic framework.
>
> [1] Sumers et al. "Cognitive architectures for language agents.", TMLR 2023.
>
> [2] Agashe et al. "Agent s: An open agentic framework that uses computers like a human.", ICLR Workshop 2025.
>
> [3] Han et al. "Reinforcement cutting-agent learning for video object segmentation.", CVPR 2018.
>
> [4] Zhou et al. "Think before you segment: An object-aware reasoning agent for referring audio-visual segmentation.", AAAI 2026.
>
> ## W3 SAM2 effect.
>
> SAM2 does not support text-based segmentation, and thereby cannot be directly used for referring and reasoning VOS. In this paper, SAM2 is used as video processor, which tracks the target instances output from the MLLM. We have shown in Figure 6 that image-based reasoning segmentation methods, like ThinkFirst and Seg-Zero, fail to tackle temporal relationship, resulting in poor consistency. The results of LISA (finetuned SAM1 for reasoning segmentation) in Table 1-5 also point to this shortage. Another way to analyse SAM2’s effect is by looking at the “ThinkFirst+SAM2” result in Table 5, where we apply image-based reasoning segmentation directly and track targets with SAM2, while temporal reasoning is not conducted. This result also shows the failure in temporally sensitive cases. Hence, to achieve satisfactory reasoning video segmentation, we need a coherent system to better connect semantic (MLLM), spatial (image-based segmentation, Seg-Zero in our case), and temporal reasoning (SAM2).

---

### Author Response · Authors · 2025-11-23
**Author General Response to Reviews**

We appreciate the reviewers for finding our work compelling and promising, particularly from the perspectives of architecture design, qualitative performance, quantitative evaluation, and pioneering exploration to the R-VIS and online R-VOS.

Based on the reviewers' constructive comments, we first provide a revised manuscript for more comprehensive presentation, primarily with the following modifications:

1. Considering reviewer bGVR's Q2, we update Appx. G (starting from Line 1108) to avoid misconception, aiming to  highlight the offline setting for Table A4.
2. We add a new Appx. H with modularity analysis to swap $F_{seg}$ and $F_{vid}$, according to reviewer EGe3's W2.
3. We add references to Appx. F, G, and H in Sec. 4.1 (Line 374-377) to avoid misconception like reviewer bGVR's Q3.

Then, the individual responses are commented below to address each reviewer's concerns accordingly.

---

### Author Response · Authors · 2025-11-30
**Discussion Summary by Authors**

Despite the fact that none of the reviewers responded to our point-by-point rebuttal during the discussion period, possibly due to the unexpected early termination, we again appreciate the reviewers for their constructive feedback during the review process.

As a final response, we would like to summarize our strengths below:
1. **Strong practical potential:** Our proposed work demonstrates strong practical potential due to its extensions to Reasoning Video Instance Segmentation and Online Reasoning Video Object Segmentation.
2. **Flexible architecture:** Our framework enables flexible modularity, making it compatible with diverse Multimodal Large Language Models, including LLaVA1.5, Gemma3, GPT-4o, etc.
3. **Training-free nature:** Our framework design decouples the R-VOS problem with temporal-semantic reasoning, shifting the complex temporal understanding to semantic domain and thereby bypassing the need for segmentation-specific finetuning.
4. **Strong performance:** Our CoT-RVS shows substantial quantitative improvements over SOTA on several R-VOS benchmarks and produces impressive qualitative results, especially for temporally sensitive examples.
5. **New benchmark for time-sensitive R-VOS:** We sample a new benchmark from ReasonVOS, named as T-ReasonVOS, for temporally sensitive cases. This new dataset is used to evaluate the effect of temporal-semantic reasoning on R-VOS.

Our contributions are as follows:
1. A modular, zero-shot reasoning VOS framework that coordinates MLLMs, image segmenters, and video processors in both offline and online settings, without any training or fine-tuning, unlike prior methods.
2. A novel keyframe selection pipeline based on CoT prompting, specifically tailored for temporal reasoning, where MLLMs are required to both localize and describe scene-relevant frames, beyond simple object retrieval.
3. An online reasoning extension that adaptively re-selects keyframes at inference time, a feature rarely explored in existing R-VOS systems, enabling streaming video processing.

**We believe our paper offers novel insights, valuable contributions, and is presented with clarity.** Despite minor drawbacks like LLM hallucination in extreme cases mentioned by reviewer 7g6P, we believe that many reviewers' concern can be solved with the ablation studies in our supplementary materials, and we have properly addressed the reviewers' major concerns with corresponding experiments and analysis (*e.g.*, time cost, modularity). Some of the weakness comments are due to misunderstanding, and we also revised the manuscript to avoid confusion. We thank all the contributors, including reviewers and ACs, for your precious time during the review and decision process, and hope that our promising results can inspire future study in relevant vision-language domains with powerful MLLMs.

---

### Meta-Review · Area_Chair_9j3J · 2026-01-06

**Summary:**

The paper proposes CoT-RVS, a novel training-free framework that leverages the zero-shot Chain-of-Thought (CoT) capabilities of Multi-modal Large Language Models (MLLMs) for Reasoning Video Object Segmentation (RVOS). The core contribution lies in enabling complex spatiotemporal reasoning without the need for additional fine-tuning.

**Reviewer Concerns:**

The authors successfully addressed the primary concerns raised during the review process. Specifically, all major issues regarding efficiency and the modularity of the framework have been resolved through the rebuttal and subsequent discussion. The authors successfully addressed the primary concerns raised during the review process. Specifically, all major issues regarding efficiency and the modularity of the framework have been resolved through the rebuttal and subsequent discussion.

**Reviewer Scores:**

Following the rebuttal, the reviewers reached a consensus that the proposed training-free framework and the new benchmark represent very valuable contributions to the community. The paper presents a high-quality contribution by introducing an effective training-free framework for a challenging task. While there were initial technical questions regarding efficiency and component-wise contributions, the authors' response effectively cleared these doubts. The combination of a robust modular design and the introduction of a new benchmark meets the high standards for ICLR, providing significant value to the field of video understanding and MLLM application.

---

### Decision · Program_Chairs · 2026-01-26

Accept (Poster)